# Diffusion-Weighted MRI and Human Papillomavirus (HPV) Status in Oropharyngeal Cancer

**DOI:** 10.3390/cancers16244284

**Published:** 2024-12-23

**Authors:** Heleen Bollen, Rüveyda Dok, Frederik De Keyzer, Sarah Deschuymer, Annouschka Laenen, Johannes Devos, Vincent Vandecaveye, Sandra Nuyts

**Affiliations:** 1Laboratory of Experimental Radiotherapy, Department of Oncology, University of Leuven, 3000 Leuven, Belgium; 2Department of Radiation Oncology, Leuven Cancer Institute, University Hospitals Leuven, 3000 Leuven, Belgium; 3Department of Radiology, University Hospitals Leuven, 3000 Leuven, Belgium; 4Leuven Biostatistics and Statistical Bioinformatics Center, University of Leuven, 3000 Leuven, Belgium

**Keywords:** oropharyngeal carcinoma, diffusion-weighted MRI, functional imaging, Human Papillomavirus (HPV)

## Abstract

This study explored how diffusion-weighted MRI (DW-MRI) differs in oropharyngeal cancer (OPC) based on HPV status before and during radiation therapy. MRI data from 178 OPC patients were analyzed and showed that HPV-positive tumors had lower initial ADC values and specific texture features than HPV-negative tumors. During treatment, the change in ADC values was higher in HPV-positive tumors compared with HPV-negative tumors. These findings highlight the potential of DW-MRI as a non-invasive biomarker for HPV status.

## 1. Introduction

Oropharyngeal squamous cell carcinoma (OPC) represents the sole subsite within head and neck cancer (HNC) with a rising incidence, a trend that has been associated with an increase in Human Papillomavirus (HPV) infections [1,2]. HPV-positive OPC exhibits a different biology compared with OPC related to tobacco and alcohol consumption [3]. Moreover, HPV-positive disease tends to have a more favorable prognosis compared with HPV-negative OPC [4]. To better reflect the prognostic difference between the two entities, the TNM classification (8th Edition) for OPC was recently revised, separating the staging for HPV positive and negative OPC [5,6]. At present, HPV status in OPC is determined by tissue sampling through p16 immunohistochemistry (IHC) as a surrogate marker and/or detection of viral HPV by Polymerase Chain Reaction (PCR) or in situ hybridization [7,8]. Magnetic resonance imaging (MRI) plays a critical role in both diagnosis and staging of OPC, as it is often the preferred imaging modality during the diagnostic work-up due to its superior soft-tissue contrast and minimal susceptibility to dental filling artifacts [9,10,11]. Diffusion-weighted (DW)-MRI is a functional imaging modality that probes tissue microstructure by evaluating the Brownian motion of water molecules. The water diffusion is quantified by the apparent diffusion coefficient (ADC), which represents the signal attenuation observed on DW-images with increasing b-values and, therefore, inversely reflects the tissue cellularity [12]. Furthermore, quantitative features beyond ADC values, obtainable through radiomics analysis of DW images, can provide further insights into the biological environment and the heterogeneity of tumors [12,13]. Previous studies have underscored the benefits of DW-MRI and ADC values to discriminate between benign and malignant lymph nodes, as well as their prognostic value in HNC [14,15,16,17,18]. Given its integration into routine clinical practice, pre-treatment DW-MRI would be a practical tool for non-invasive detection of HPV status. To date, however, the ability of DW-MRI as a non-invasive biomarker to differentiate HPV status in OPC remains ambiguous.

The aim of this prospective study is to investigate the association of quantitative DW-MRI imaging features and HPV status in OPC. Given that HPV-positive and negative OPC are recognized as distinct clinical entities with differences in histological features, we hypothesize that these tumors will exhibit different parameters on pre-treatment DW-MRI [19]. In addition, considering the differential response to radiation treatment (RT), with HPV-positive OPC demonstrating greater radiosensitivity, we hypothesize that this dissimilarity will be quantifiable through ADC parameters during RT.

## 2. Materials and Methods

### 2.1. Patient and Treatment Characteristics

A cohort of 178 patients diagnosed with histologically confirmed OPC was prospectively included between 2005 and 2018. All patients were treated with primary RT, chemoradiotherapy (CRT) or RT with cetuximab. An equivalent RT dose of 70 Gy in fractions of 2 Gy was administered. Chemotherapy and cetuximab regimens entailed three weekly intravenous cisplatin at a dosage of 100 mg/m^2^ or weekly intravenous cetuximab at a dosage of 400 mg/m^2^. Treatment decisions were guided by a multidisciplinary tumor board. Pre-treatment work-up included DW-MRI, with additional DW-MRI conducted during the 4th week of RT. Subsequent follow-up appointments were scheduled every two, three, four and six months in the first two years, third, fourth and fifth years, respectively. During follow-up, a full clinical examination with laryngoscopy was routinely performed. Imaging (MRI, CT or FDG-PET/CT) was conducted 12 weeks post-RT completion and subsequently whenever there was a clinical suspicion of recurrence. This study received approval from the Ethics Committee of the University Hospitals of Leuven, and all participants provided written informed consent (NTC01829646).

### 2.2. HPV Status

HPV status was assessed using biopsy samples obtained during diagnostic work-up. Samples were classified as HPV positive if p16 IHC revealed more than 70% cytoplasmic and nuclear staining [20]. Biopsy samples collected before 2011 underwent retrospective review, while p16 staining was prospectively conducted on samples acquired after 2011.

### 2.3. MRI Imaging Protocol and Image Data Analysis

Each patient underwent an MRI prior to RT, as well as during the 4th week of RT. MRI examinations were conducted initially using a 1.5 Tesla (T) system (Sonata; Siemens, Erlangen, Germany) until 2010, transitioning to a 3 T system (Achieva, Philips, Best, The Netherlands) from 2010 onwards, equipped with dedicated phased-array head and neck receiver coils. For anatomical correlation, a transverse T2-weighted turbospin-echo (TSE) sequence was employed, featuring a repetition time (TR)/echo time (TE) of 3080/106 ms, a field of view (FoV) measuring 203 × 250 mm^2^, a matrix of 291 × 512, 48 slices with a slice thickness of 4 mm, a 0.4 mm intersection gap and 2 averages acquired. This configuration resulted in an in-plane resolution of 0.7 × 0.5 mm^2^, with an acquisition time of 5 min and 42 s. Additionally, a transverse DW-MRI sequence was obtained, encompassing 6 b-values (0, 50, 100, 500, 750 and 1000 s/mm^2^), with a TR/TE of 7400/84 ms, a FoV of 203 × 250 mm^2^, a matrix of 104 × 128, 48 slices with a slice thickness of 4 mm, a 0.4 mm intersection gap and 3 averages acquired. This configuration yielded an in-plane resolution of 2 × 2 mm^2^, with an acquisition time of 6 min and 3 s.

The region of interest (ROI) was delineated by a radiologist with over ten years of experience in both pre-treatment MRI scans and MRIs acquired during RT. Delineation was performed on the ADC map, utilizing a semi-automated constrained region growing technique using visual comparison to the original b-value images. Calculation of ADC maps was performed using the mono-exponential model employing all acquired b-values. All delineations were carried out without knowledge of the HPV status of the tumor. The ROI encompassed the entire volume of the primary tumor. The primary tumor volume of all oropharyngeal tumors was measured using the RT planning system (Varian Medical Systems, Palo Alto, CA, USA), and encompassed the entire pre-treatment gross tumor volume.

A total of 105 radiomic parameters were computed and averaged over the DWI-MRI images using the open-source PyRadiomics software (Version 1.3). The parameters included 13 shape features, 18 first-order parameters, 23 Gray Level co-occurrence matrix (GLCM) parameters, 14 Gray Level Dependence Matrix (GLDM) parameters, 16 Gray Level Run Length Matrix (GLRLM) parameters, 16 Gray Level Size Zone Matrix (GLSZM) parameters and 5 Neighboring Gray Tone Difference Matrix (NGTDM) parameters. For the first-order histogram parameters, the 10th percentile, 90th percentile, minimum, maximum, mean and median were calculated for the MRI scan taken at the start and in the fourth week of the treatment. The delta (Δ) ADC values, the percentage of ADC changes between pre-treatment and during RT, were calculated using the formula: ΔADC = [(ADC_during_ − ADC_pre_)/ADC_pre_] × 100.

### 2.4. Statistics

Patient, tumor and treatment characteristics were compared with two-sided Fisher’s exact test for categorical variables and the Mann–Whitney U test for continuous variables. All ADC parameters and 105 texture parameters were compared between HPV positive and negative OPC using the Mann–Whitney U test. The Spearman correlation coefficient was used to estimate the association between two continuous variables. The Kaplan–Meier method was used for presenting oncological outcomes. Overall survival (OS) was calculated from the first day of RT to the date of death from any cause. Locoregional control (LRC) was calculated from the first day of RT to the date of recurrence. Patients on follow-up and patients lost to follow-up were censored at the last date at which they were known to be alive (OS) or recurrence-free (LRC). The association between first-order or texture parameters and oncologic outcomes was examined using Cox regression models, applying logarithmic transformation for continuous predictors to model non-linear associations. Results are presented as hazard ratios for a 2-fold increase of the predictor, with 95% confidence intervals. A forward stepwise selection procedure was used to develop a multivariable logistic regression model for the prediction of HPV-positive status. The significance threshold was set at a *p*-value of <0.05. All tests were performed at a two-sided 5% significance level. No correction for multiplicity was performed due to the exploratory nature of the study. Analyses have been performed using SAS software (version 9.4 of the SAS System for Windows). Boxplots were generated using Graphpad (version 10).

## 3. Results

### 3.1. Patient, Tumor and Treatment Characteristics

A total of 178 patients were initially included, of which 158 patients had known p16 status, available pre-treatment DW-MRI and DW-MRI during treatment, and thus were withheld for further analysis (Figure 1). Patient, tumor and treatment characteristics are summarized in Table 1. Primary tumor locations included tonsils (n = 70, 39%), the base of tongue (n = 57, 32%), posterior pharyngeal wall (n = 22, 12%), soft palate (n = 11, 6%), vallecula (n = 13, 7%) and lateral pharyngeal wall (n = 5, 3%). A total of 60 out of 158 OPC patients (38%) were considered HPV positive. No significant differences regarding tumor stage, gender, age or tumor volume were observed between HPV-negative and -positive tumors. Patients with HPV-negative tumors showed significantly more smoking pack years and alcohol consumption. HPV-positive tumors were located more often in the tongue base and tonsils (n = 23, 38.33% and 30, 50.0%). The proportion of patients with HPV-positive tumors was significantly higher in the subgroup scanned at 3T MRI in comparison to 1.5 T. The average number of days between the first and second MRI was 28 days.

### 3.2. First-Order Parameters

DW-MRI first-order parameters according to HPV status are presented in Table 2. Pre-treatment first-order 10th percentile ADC was significantly lower in HPV-positive compared with HPV-negative tumors (78.75 × 10^−5^ mm^2^/s vs. 86.45 × 10^−5^ mm^2^/s, respectively, *p* = 0.03). The remaining pre-treatment first-order parameters showed no significant differences, although a trend towards reduced mean and median ADC values in the HPV-positive subgroup was observed with a mean ADC value of 119.88 × 10^−5^ mm^2^/s for HPV-negative and 107.47 × 10^−5^ mm^2^/s for HPV-positive OPC (Figure 2A and Table 2). During the fourth week of RT, the mean, median, minimum, 10th percentile, 90th percentile and maximum ADC values were significantly higher in the HPV-positive OPC compared with HPV-negative OPC (Figure 2B and Table 2). The Δ ADC mean was significantly higher for the HPV-positive OPC group (mean value of 95% for HPV-positive vs. 55% for HPV-negative, *p* < 0.01) (Figure 2C).

### 3.3. Texture Analysis and Predictive Value

Full texture analysis can be found in Appendix A. GLCM correlation was the only texture parameter that significantly differed between the two entities, with a median value of 0.18 and 0.30 for HPV-positive and -negative OPC, respectively (*p* < 0.01). A predictive model based on clinical factors, including smoking status, alcohol consumption, tumor location and radiomic features, including 10th percentile ADC, mean ADC and GLCM correlation, yielded an area under the curve of 0.77 (95% CI 0.70–0.84) (Figure 3).

### 3.4. Oncologic Outcome

The median follow-up time was 5.1 years. Kaplan–Meier curves for LRC and OS are shown in Figure 4A,B. The estimated 2- and 5-year LRC was 80% (93% for HPV positive vs. 71% for HPV negative, *p* < 0.01) and 75% (89% for HPV positive vs. 64% for HPV negative, *p* < 0.01), respectively. The 5-year OS was 55% (75% for HPV positive vs. 46% for HPV negative, *p* < 0.01). All pre-treatment ADC values were significantly higher in patients who developed locoregional recurrence (LRR) after treatment, with a hazard ratio (HR) of 1.85 for the ADC mean (*p* < 0.01), while ADC during treatment and ΔADC were significantly lower (Appendix A).

## 4. Discussion

Because HPV is widely recognized as a key prognostic biomarker in OPC patients, routine testing for HPV is strongly recommended [21]. The latest guidelines from the American Joint Committee on Cancer (AJCC) classify HPV-positive and -negative OPC as distinct clinical entities, and several clinical trials are focusing on treatment de-escalation for HPV-positive patients [22,23]. The increased clinical emphasis on HPV status has driven researchers to develop and validate accessible surrogate markers for reliably identifying HPV-related OPC [2]. While traditional HPV analysis relies on tissue samples, radiomic-based phenotyping—encompassing intensity, shape and texture analysis from medical imaging—has emerged as a promising tool to distinguish between HPV-positive and -negative OPC. Radiomics features are quantitatively derived through mathematical algorithms, capturing valuable information that is not immediately visible to radiologists. These features hold the potential to serve as a form of virtual biopsy, as variations in texture might reveal underlying microstructural differences and unique pathological patterns within a lesion. Current evidence supporting the role of radiomic features in identifying HPV status in OPC is limited [24,25,26,27]. Although a few studies have shown promising potential, most of these studies rely on computed tomography (CT) images, owing to their widespread availability, ease of data extraction, and standardization across different scanners [24]. However, many institutions incorporate MRI into the clinical workflow for OPC treatment due to its superior soft tissue contrast and its ability to provide insights into the functional properties of tumors [10,11]. This is particularly important for radiomic phenotyping, as HPV-positive OPC histologically differs from HPV-negative OPC [19,28]. The use of ADC values as prognostic indicators, as well as for assessing treatment response and tumor characterization, has been reported for various cancer types [12,29,30,31]. However, the use of ADC maps for extracting and analyzing radiomic features for OPC has been explored in only a few studies [29,31,32,33].

The HPV-positive disease is characterized by a non-keratinizing morphology, low stromal volume and large amounts of tumor-infiltrating lymphocytes. Several studies have demonstrated that mean ADC values are inversely correlated with cell density and positively correlated with the percentage of stroma, emphasizing the importance of MRI in evaluating these parameters [29,32,34]. While most studies indicate that the mean ADC values are significantly lower in HPV-positive OPC compared with HPV-negative OPC [29,33,35,36], other trials have found no significant differences between the two clinical entities [37,38]. Our results show lower mean, medium, minimum, maximum, 90th and 10th percentile ADC values in HPV-positive disease, confirming the trend for lower diffusion coefficients and, hence, increased cellularity and reduced stromal volume in HPV-positive OPC. Our findings support those of Marzi et al., who attributed the lower ADC in HPV-positive disease to ADC histograms that exhibited greater leptokurtosis and skewness, indicating increased cellularity in HPV-positive OPC [39]. Previous radiomic studies, predominantly utilizing CT imaging modalities, have reported HPV-positive OPC to be characterized by spherical and distinctly defined margins [24,30,40]. Although our study did not reveal significant disparities in shape parameters, we observed lower volume and surface area parameters in the HPV-positive cohort. Notably, GLCM correlation was the sole texture parameter that differed significantly between the two entities, with HPV-positive OPC demonstrating a lower median value compared to HPV-negative OPC. GLCM features, which quantify the frequency of pixel pair occurrences, offer valuable insights into the linear relationship between intensity levels within tissues. Lower GLCM correlation values indicate heightened randomness or heterogeneity in the spatial distribution of intensity levels within localized regions [25,27,41,42]. Moreover, HPV-positive OPC has been associated with patchier CT images, while elevated GLCM correlation has been linked to more aggressive subtypes using FDG PET-based radiomic techniques [24,27,42,43].

A model to predict the HPV status based on clinical factors, including smoking status, alcohol consumption, tumor location and radiomic features, including 10th percentile ADC, mean ADC and GLCM correlation, yielded an area under the curve of 0.770. While this indicates an acceptable level of discriminatory performance, the predictive power of DW-MRI is not yet near the one of pathological confirmation [7,8]. Therefore, despite its theoretical validity, the current role of DW-MRI remains supplementary. Although the presented results do not yet suggest a substitutive role for radiomics features instead of routine practices such as IHC, the established features offer additional insights that can complement the current conventional methods. Moreover, the use of artificial intelligence and deep learning algorithms, along with the accessibility of larger datasets that integrate existing biological and pathological data, will likely enhance the accuracy and reliability of radiomic models for HPV discrimination. This advancement will probably extend beyond determining HPV status to predicting prognosis and treatment responses, regardless of the tumor’s HPV status. Further research should, therefore, make use of existing public datasets to validate all published features, considering clinical factors such as tumor staging when building predictive models. Additionally, further research should explore the potential correlations among various MRI characteristics to enhance our understanding of the underlying mechanisms behind the observed findings.

During RT, the enhanced treatment response observed in HPV-positive OPC was quantifiable through higher mean ADC and ΔADC values compared with the HPV-negative subgroup. These findings confirm previous research demonstrating the prognostic value of DW-MRI parameters [14,16,44,45]. The information gathered from delta radiomics could, in the future, be used for decision-making regarding treatment adaptation or other interventions beneficial to the patient [46].

The strength of this study lies in the inclusion of a large, prospective patient cohort and the utilization of the widely accessible open-source PyRadiomics software. An extensive meta-analysis and systemic review, including 26 studies, highlighted the overall low quality of existing radiomics investigations, thereby constraining the generalizability of their findings [24]. Initiatives such as the Image Biomarker Standardization Initiative (IBSI) aim to standardize the extraction of image biomarkers from medical imaging data [25,47]. Leveraging PyRadiomics, which is largely compliant with the IBSI initiative, ensures standardized extraction and analysis of radiomic features, enhancing the reproducibility of our findings across diverse studies. Moreover, all existing studies reporting on ADC values, radiomics and HPV status had a retrospective study design and included a smaller number of patients. In addition, we opted to delineate the entire volume on the ADC map using a semi-automated balloon inflation technique. This approach is in contrast to other approaches, ranging from including only a single section with the largest diameter to manually delineating the entire 3D volume [29,34,35,36,37,48,49,50], and offers the advantage of reducing inter-observer variability compared with delineating only one tumor section and allows us to examine the patterns in pixel intensities for the texture analysis.

The limitations of our study include the sole use of p16 immunostaining as a surrogate marker for HPV, the absence of a validation cohort, the absence of an evaluation of lymph nodes and the use of different MRI field strengths. Concerning the latter, a larger proportion of HPV-positive OPC cases received a 3T MRI, which can be explained by the examination year and the rising incidence of HPV-positive OPC over the past decade [51]. Given that the scan protocol was adjusted for the two MRI settings, we anticipate minimal impact on our outcomes. An exploratory subgroup analysis was performed, which revealed no disparity in results in ADC mean values and texture parameters between the 1.5 T and 3 T subgroups. Another limitation of this study is that the ROI was defined by a single radiation oncologist despite their extensive experience of over 15 years. While the expertise of this clinician adds credibility to the definition of the ROI, the lack of multiple reviewers introduces potential subjectivity and variability in the delineation process. Future studies could benefit from involving multiple radiation oncologists to reduce bias and ensure greater consistency in the definition of the ROI.

## 5. Conclusions

Our results highlight the potential of DW-MRI imaging as a non-invasive imaging biomarker for the prediction of HPV status, although radiomics features derived from imaging are currently less accurate than the well-established IHC methods. The use of advanced machine learning techniques and implementation of radiomic features with clinical, histological and biological data will enhance the ability of radiomic analysis to determine the HPV status.

## Figures and Tables

**Figure 1 cancers-16-04284-f001:**
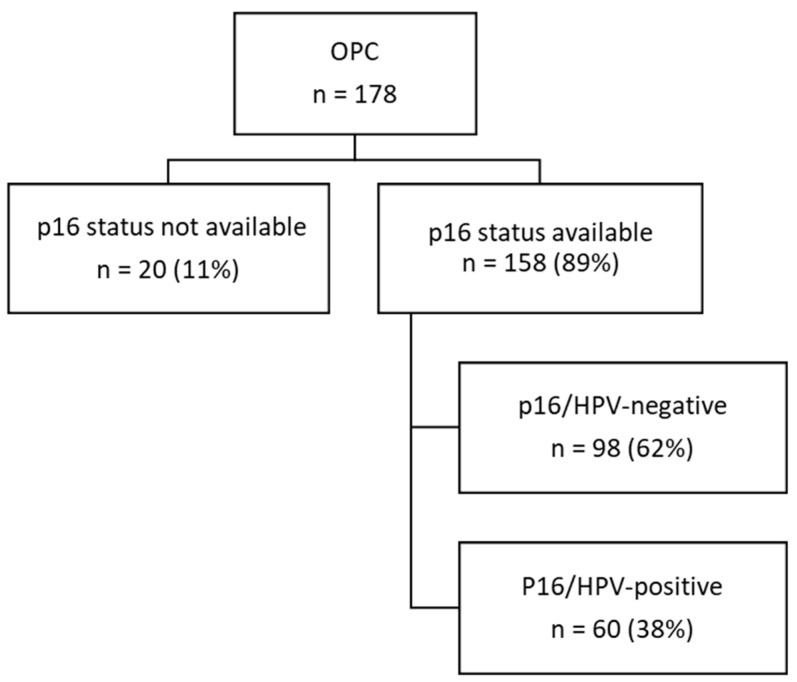
Flowchart of the study. OPC: oropharyngeal cancer; HPV: Human Papillomavirus, n = number of patients. P16 was used as surrogate marker for HPV. In the following figures and tables, p16-positive tumors will be depicted as HPV-positive and p16-negative tumors as HPV-negative.

**Figure 2 cancers-16-04284-f002:**
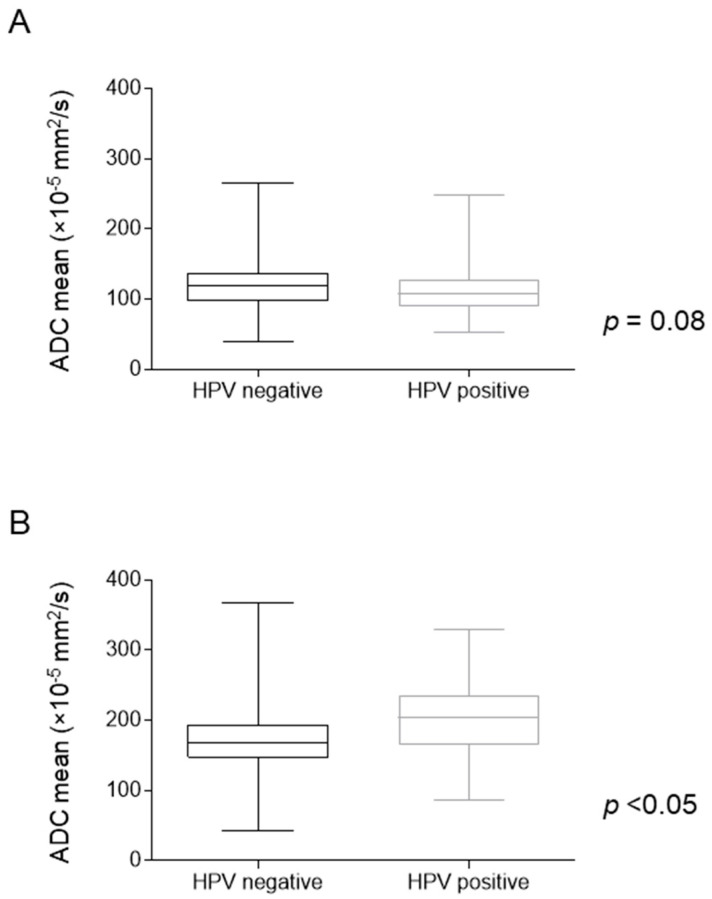
Boxplots displaying the distribution of (**A**) pre-treatment ADC mean values (×10^−5^ mm^2^/s), (**B**) ADC mean values during RT (×10^−5^ mm^2^/s) and (**C**) ΔADC mean (%) according to HPV status. *p*-values were calculated with Mann–Whitney *U* test.

**Figure 3 cancers-16-04284-f003:**
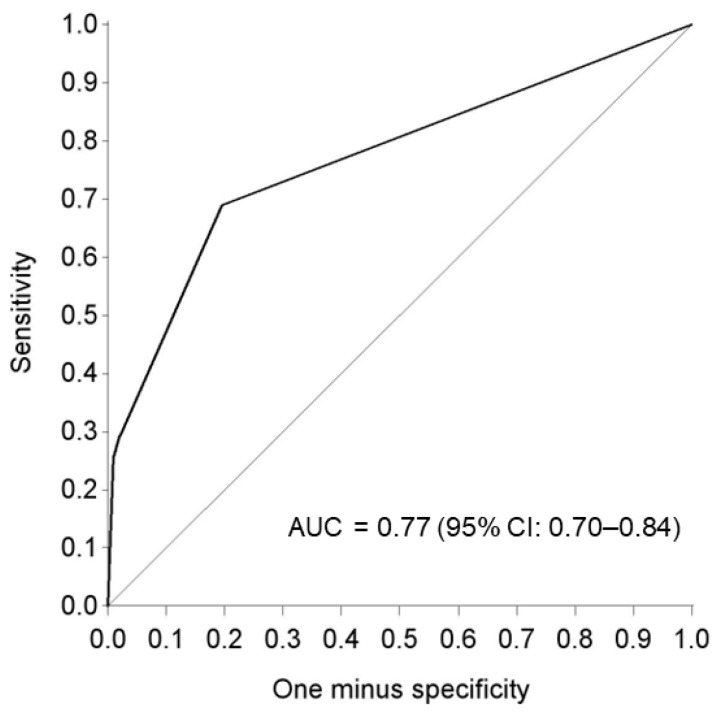
Receiver operating characteristic (ROC) curve of the predictive model based on clinical factors, including smoking status, alcohol consumption, tumor location and radiomic features, including 10th percentile ADC, mean ADC and GLCM-correlation. HPV status was used as classification variable. The predicted probability was generated by multivariable logistic regression. AUC: Area under the curve.

**Figure 4 cancers-16-04284-f004:**
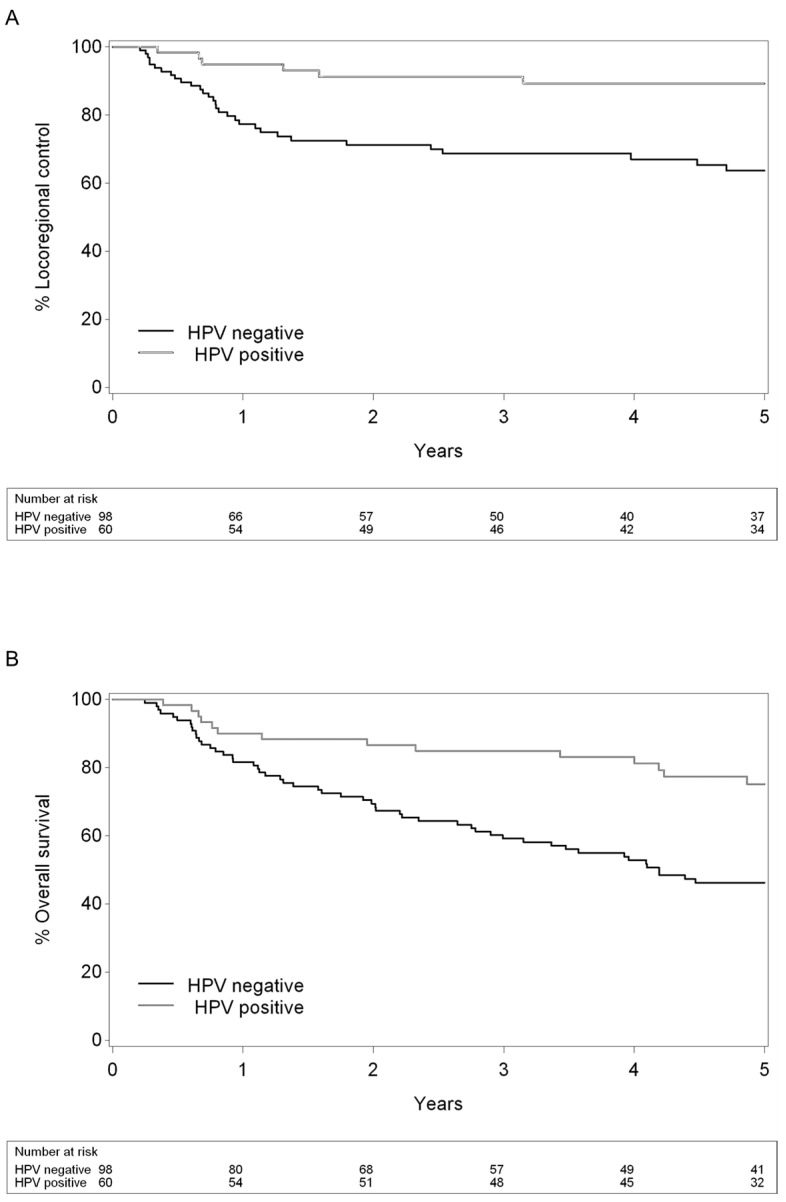
Kaplan–Meier curves for (**A**) locoregional control (LRC) and (**B**) overall survival (OS) stratified by p16 status as a surrogate marker for HPV. HPV: Human Papillomavirus.

**Table 1 cancers-16-04284-t001:** Patient, tumor and treatment characteristics.

Variable	HPV Negative	HPV Positive	*p*-Value
Gender (n, %)	98	60	
Male	86 (88)	49 (82)	0.35
Female	12 (12)	11 (18)	
Age at the start of RT (n)	98	60	0.41
Mean (std)	59.12 (11.13)	60.61(10.11)	
Median	58.09	61.07	
T-classification (n, %)	98	60	
T1	4 (4)	5 (8)	0.05
T2	31 (32)	30 (50)	
T3	27 (27)	11 (19)	
T4a	29 (30)	9 (15)	
T4b	7 (7)	5 (8)	
N-classification (n, %)	98	60	
N0	15 (15)	10 (17)	0.33
N1	21 (21)	8 (13)	
N2a	4 (4)	2 (3)	
N2b	23 (24)	23 (39)	
N2c	32 (33)	14 (23)	
N3	3 (3)	3 (5)	
AJCC stage 7th TNM edition (n, %)	98	60	
I	0 (0)	1 (2)	0.44
II	7 (7)	5 (8)	
III	20 (21)	8 (13)	
IVA	63 (64)	38 (63)	
IVB	8 (8)	8 (13)	
Chemotherapy (n, %)	98	60	
No	25 (26)	12 (20)	0.18
Platinum-based	63 (64)	46 (77)	
Targeted	10 (10)	2 (3)	
Smoking (Packyears) (n)	89	50	<0.01
Mean (std)	35.58 (16.32)	21.00 (21.01)	
Median	35.00	20.00	
Tumor volume (cc) (n)	91	53	0.12
Mean (std)	28.23 (29.25)	19.75 (14.42)	
Median	19.40	16.10	
IQR	(11.30; 32.80)	(10.30; 24.90)	
Range	(1.40; 192.90)	(2.30; 74.60)	
Alcohol (n, %)	97	58	
Active heavy drinker	54 (56)	16 (28)	<0.01
Never	3 (3)	10 (17)	
Occasional	15 (15)	28 (48)	
Past drinker	25 (26)	4 (7)	
Location (n, %)	98	60	
Posterior wall	14 (14)	4 (7)	0.03
Lateral wall	4 (4)	0 (0)	
Soft palate	8 (8)	1 (2)	
Tongue base	25 (26)	23 (38)	
Tonsil	37 (38)	30 (50)	
Vallecula	10 (10)	2 (3)	
MRI (n, %)	98	60	<0.01
1.5 T	50 (51)	23 (38)	
3T	48 (49)	37 (62)	

**Table 2 cancers-16-04284-t002:** First-order parameters according to HPV status before and during treatment.

	Start of RT			Week 4 of RT		
	HPV Negative	HPV Positive	*p*-Value	HPV Negative	HPV Positive	*p*-Value
First Order Parameters	Median [IQR]	Median [IQR]		Median [IQR]	Median [IQR]	
original_firstorder_10Percentile	86.45 [72.00;105.00]	78.75 [67.50;90.00]	0.03	132.80 [114.20;161.00]	170.50 [135.15;197.00]	<0.01
original_firstorder_90Percentile	149.00 [118.90;175.00]	142.00 [111.50;163.20]	0.21	204.85 [175.00;241.70]	239.00 [199.60;282.70]	0.01
original_firstorder_Energy	6.33 × 10^6^ [1.81 × 10^6^;2.23 × 10^7^]	3.75 × 10^6^ [1.32 × 10^6^;1.66 × 10^7^]	0.45			
original_firstorder_Entropy	1.97 [1.64;2.34]	1.95 [1.52;2.37]	0.67			
original_firstorder_InterquartileRange	29.13 [22.00;41.00]	30.13 [20.63;40.00]	0.68			
original_firstorder_Kurtosis	3.19 [2.84;3.84]	3.16 [2.69;3.91]	0.45			
original_firstorder_Maximum	196.50 [146.00;241.00]	169.00 [145.50;255.50]	0.41	231.50 [202.00;281.00]	263.00 [218.50;328.00]	0.03
original_firstorder_MeanAbsoluteDeviation	18.34 [14.41;25.04]	18.12 [12.689;24.748]	0.63			
original_firstorder_Mean	119.88 [96.93;137.50]	107.47 [89.68;126.49]	0.08	168.22 [146.22;193.65]	203.79 [165.86;234.87]	<0.05
original_firstorder_Median	116.00 [95.50;137.00]	105.00 [88.75;124.25]	0.06	168.75 [145.75;195.25]	203.25 [164.50;233.50]	<0.05
original_firstorder_Minimum	48.50 [22.00;63.00]	39.00 [3.50;56.50]	0.22	102.00 [65.00;138.50]	144.50 [100.50;167.00]	<0.05
original_firstorder_Range	139.50 [107.00;207.00]	144.50 [94.00;230.00]	0.56			
original_firstorder_RobustMeanAbsoluteDeviation	12.19 [9.70;16.70]	13.14 [9.02;16.43]	0.75			
original_firstorder_RobustMeanAbsoluteDeviation _1	122.02 [99.36;143.30]	110.53 [91.31;129.38]	0.10			
original_firstorder_Skewness	0.05 [−0.33;0.35]	0.15 [−0.10;0.38]	0.23			
original_firstorder_TotalEnergy	6.33 × 10^6^ [1.81 × 10^6^;2.23 × 10^7^]	3.75 × 10^6^ [1.32 × 10^6^;1.66 × 10^7^]	0.45			
original_firstorder_Uniformity	0.31 [0.23;0.38]	0.30 [0.23;0.42]	0.80			
original_firstorder_Variance	533.75 [334.86;982.88]	535.49 [260.10;1020.70]	0.57			

IQR = [Q1; Q3], Q1: first quartile, Q3: third quartile; *p*-value from Mann–Whitney *U* test.

## Data Availability

The authors do not own these data and hence are not permitted to share them in the original form (only in aggregate form, e.g., publications). Applications are reviewed and approvals granted subject to meeting all ethical and research conditions set forth by the Ethics Committee Research UZ/KU Leuven.

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
