# Peer review of "Diffusion-Weighted MRI and Human Papillomavirus (HPV) Status in Oropharyngeal Cancer"

_cancers, 2024, doi:10.3390/cancers16244284_

Round 1
Reviewer 1 Report
Comments and Suggestions for Authors
1. Is p16 IHC used to assess HPV status sufficiently sensitive for HPV definitive identification? Were PCR or immunohistochemical studies performed and how did they correlate with p16 IHC test?
2. Is it possible to compare MRI examinations performed using 1.5 Tesla and 3 Tesla system machines?
3. Was the Region of Interest (ROI) evaluated by the same radiologist in all cases?
4. <...One-hundred seventy-eight patients were initially included, of which 158 patients (163) had known p16 status...> And what about others?
5. Smoking (Packyears) (n) 151= 89+50 ? (Table 1) Calculation?
6. T-classification (n, %) 178= 98+ 60 ? (Table 1) Check calculation?
7. I agree that study limitations include the sole use of p16 immunostaining as a surrogate. The question is what criteria are associated HPV-positive OPC cases with 3T MRI performed, how do they correlate?
8. In your study, you show that the etiology of positive HPV is a diagnostic and prognostic criterion. And what then with HPV negatives, how is the disease diagnosed, and the cause remains idiopathic? As far as I understand, the treatment tactics do not change or are there any therapeutic recommendations?
Author Response
We thank all the reviewers for their constructive comments on our manuscript. We appreciate the feedback and believe it enhanced the quality of our manuscript. We have carefully considered each comment and added point by point responses below.
Reviewer 1
- Is p16 IHC used to assess HPV status sufficiently sensitive for HPV definitive identification? Were PCR or immunohistochemical studies performed and how did they correlate with p16 IHC test?
This is indeed a valid and important consideration. We acknowledge that in our study, PCR correlation was performed for only 30% of p16-positive patients, which is a limitation of our study. We agree that p16 IHC alone may not always be sufficiently sensitive for definitive HPV identification. We are also aware that international guidelines now recommend confirming HPV status through DNA based testing. We have addressed this limitation in the discussion section of our revised manuscript (line 316, highlighted in blue).
- Is it possible to compare MRI examinations performed using 1.5 Tesla and 3 Tesla system machines?
Thank you for the valuable suggestion. We have had several discussions with our radiology department regarding this matter. This is also the reason we conducted a subgroup analysis, which showed no significant differences in the mean ADC values or texture parameters between the 1.5 T and 3 T subgroups. We have elaborated on this in the discussion (line 318-324, highlighted in blue).
Furthermore, we anticipate minimal impact on our results due to the use of ADC mapping, which is more robust than conventional imaging. The radiology team has confirmed that the scan protocol was adjusted for both MRI settings, which should help minimize any potential effect on our findings. However, we do acknowledge that this remains a limitation of the study, as mentioned in the limitations section.
- Was the Region of Interest (ROI) evaluated by the same radiologist in all cases?
Indeed, the delineation was performed by a single radiologist; however, this radiologist has over 15 years of experience in head and neck cancer and extensive expertise in delineation on diffusion-weighted MRI. Nevertheless, we have acknowledged this as a limitation of the study in the discussion (lines 324-329).
- <...One-hundred seventy-eight patients were initially included, of which 158 patients (163) had known p16 status...> And what about others?
Following the guidance of our statistician, the twenty patients without known p16 status were excluded from the definitive analyses regarding the correlation of HPV and ADC. This also clarifies questions 5 and 6: in response to question 5, smoking status was unavailable for all patients. However, we realize that this may be confusing for the reader. To enhance clarity, we have revised Table 1 to display the distribution of patient characteristics exclusively for HPV-negative and HPV-positive groups (highlighted in blue in Table 1).
- Smoking (Packyears) (n) 151= 89+50 ? (Table 1)
See revised Table 1
- T-classification (n, %) 178= 98+ 60 ? (Table 1)
See revised Table 1
- I agree that study limitations include the sole use of p16 immunostaining as a surrogate. The question is what criteria are associated HPV-positive OPC cases with 3T MRI performed, how do they correlate?
A larger proportion of HPV-positive OPC cases underwent a 3T MRI. This can be attributed to the examination year and the increasing incidence of HPV-positive OPC over the past decade, which represents a limitation of the study. However, to address this, we conducted a subgroup analysis, which revealed no significant differences in mean ADC values or texture parameters between the 1.5T and 3T subgroups. Therefore, we believe the impact of this limitation is minimal, particularly since we used ADC mapping, which provides a more robust measurement. This was also mentioned more elaborately in the discussion (highlighted in blue, line 321-323).
- In your study, you show that the etiology of positive HPV is a diagnostic and prognostic criterion. And what then with HPV negatives, how is the disease diagnosed, and the cause remains idiopathic? As far as I understand, the treatment tactics do not change or are there any therapeutic recommendations?
HPV-negative disease is currently diagnosed similarly to HPV-positive disease, initially through p16 IHC, followed by DNA detection if p16 is positive. This diagnostic approach helps differentiate between HPV-related and non-related tumors. The reviewer is right in his comment that treatment strategies do not currently rely on HPV status; however, we are confident that this will change in the near future. Several studies are investigating the safety of dose de-escalation in HPV-positive disease and dose escalation in HPV-negative disease. We anticipate that these studies will provide important insights soon, and some guidelines are already beginning to differentiate treatment based on HPV status. For example, there are emerging recommendations to omit unilateral neck radiation in certain cases of HPV positive disease, which is not recommended in case of HPV negative disease. Furthermore, information regarding HPV status is crucial for staging, as HPV-positive and negative tumors are staged differently and follow different post-treatment imaging protocols. While it is true that, at this moment, treatment remains primarily based on prognostic information, we expect these findings to soon influence clinical decisions. An easy, non-invasive method for identifying HPV-positive disease would undoubtedly be a valuable tool for clinical practice.

Reviewer 2 Report
Comments and Suggestions for Authors
On the whole, this is an interesting and adequately done study on diffusion-weighted MR imaging in oncology with promising results. However, the manuscript contains several minor issues that I shall list as follows:
Abstract, P1, l32 – Please rewrite the „5“ and „2“ exponents in superscript form and add a minus before „5“ the first time it appears in the text, i.e., replace „10-5 m2/s“ with „10-5 m2/s.
Abstract, P1, l38 – Remove „I“ or „imaging“ from „DW-MRI imaging“.
Abstract, P1, l38 – „imaging“ after „non-invasive“ is superfluous.
Introduction, P2, l52 – „betweeen“ would in my opinion be a better expression than „among“.
Introduction, P2, l56 – In „high-risk HPV“, did you mistype „OPC“ as „HPV“? If not, the sentence is confusing.
Introduction, P2, l79 – „parameters“ after „values“ is unnecessary.
Materials and Methods, P2, l86 – Please insert „at a dosage of“ before „400“.
Materials and Methods, P3, l119 – „a“ should be added between „by“ and „radiologist“.
Results, P4, l174 – „scanned at“ would be a more fitting expression than „receiving“.
Discussion – The fact that only one radiologist performed the ADC measurements is another weak point of the study that should be added to the „limitations“ paragraph.
Author Response
We thank all the reviewers for their constructive comments on our manuscript. We appreciate the feedback and believe it enhanced the quality of our manuscript. We have carefully considered each comment and added point by point responses below.
Reviewer 2
On the whole, this is an interesting and adequately done study on diffusion-weighted MR imaging in oncology with promising results. However, the manuscript contains several minor issues that I shall list as follows:
Abstract, P1, l32 – Please rewrite the „5“ and „2“ exponents in superscript form and add a minus before „5“ the first time it appears in the text, i.e., replace „10-5 m2/s“ with „10-5 m2/s.
Thanks to the reviewer for this comment. This was an oversight on our part, and we have since made the necessary changes. The changes are highlighted in blue.
Abstract, P1, l38 – Remove „I“ or „imaging“ from „DW-MRI imaging“.
This was adjusted according to the reviewer suggestion. The changes are highlighted in blue.
Abstract, P1, l38 – „imaging“ after „non-invasive“ is superfluous.
This was adjusted according to the reviewer suggestion. Changes are highlighted in blue.
Introduction, P2, l52 – „between“ would in my opinion be a better expression than „among“. We agree. This was adapted according to the reviewer suggestion. Changes are highlighted in blue.
Introduction, P2, l56 – In „high-risk HPV“, did you mistype „OPC“ as „HPV“? If not, the sentence is confusing.
We agree with the reviewer. „High-risk“ was omitted in the sentence. Changes are highlighted in blue.
Introduction, P2, l79 – „parameters“ after „values“ is unnecessary.
This was adjusted according to the reviewer suggestion. Changes are highlighted in blue.
Materials and Methods, P2, l86 – Please insert „at a dosage of“ before „400“.
This was adjusted according to the reviewer suggestion. Changes are highlighted in blue, line 84-85.
Materials and Methods, P3, l119 – „a“ should be added between „by“ and „radiologist“.
This was adjusted according to the reviewer suggestion. Changes are highlighted in blue, line 117.
Results, P4, l174 – „scanned at“ would be a more fitting expression than „receiving“.
This was adapted according to the reviewer suggestion. Changes are highlighted in blue, line 171.
Discussion – The fact that only one radiologist performed the ADC measurements is another weak point of the study that should be added to the „limitations“ paragraph.
We agree with the reviewer's comment. We added a section about this limitation in the discussion: line 325-329.

Reviewer 3 Report
Comments and Suggestions for Authors
Dear Authors,
You wrote a well-structured, relevant scientific article. As You acknowledge at the end, it may not change much in clinical practice , but it is also a very pertinent knowledge for cancer society. And I respect and appreciate the very thourough acknowledgment in the Discussion session (lines 286-296) You make about a plausible role of MRI features alone in the future
Concerning some special comments:
Methods and Materials:
Lines 85-87: You describe two different systemic treatment remedies. Do You have some thoughts or is there any knowledge if that could have influenced changes in MRI features You analysed?
Lines 89-91: Is that very close follow-up is the standard in the whole country?
The features that You chose to analyze in image data analyses, why did You choose all of them? And further on that: in lines 222-225 You describe the differences of ADC values. Can there be a cut-off or a specific threshold of prognostic ADC value for all OPC patients? That is a general question..
What would the sensitivity or specificity of ADC values be for treatment responces for OPC? Maybe not from Your data, but do you have any knowledge on that?
The reference list is to a heavy side...
Best regards
Author Response
We thank all the reviewers for their constructive comments on our manuscript. We appreciate the feedback and believe it enhanced the quality of our manuscript. We have carefully considered each comment and added point by point responses below.
Reviewer 3
You wrote a well-structured, relevant scientific article. As You acknowledge at the end, it may not change much in clinical practice , but it is also a very pertinent knowledge for cancer society. And I respect and appreciate the very thorough acknowledgment in the Discussion session (lines 286-296) You make about a plausible role of MRI features alone in the future
Concerning some special comments:
Methods and Materials:
Lines 85-87: You describe two different systemic treatment remedies. Do You have some thoughts or is there any knowledge if that could have influenced changes in MRI features You analyzed?
We indeed two different systemic treatments: Cisplatin and Cetuximab. To elaborate, at the time of the study, patients who could not receive chemotherapy were often treated with cetuximab, which is a monoclonal antibody that targets the epidermal growth factor receptor (EGFR). Although cetuximab was used as an alternative to chemotherapy, it was eventually phased out due to its inferior outcomes compared to chemotherapy, as well as its relatively high toxicity profile. These shortcomings led to its discontinuation as a standard regimen for many patients. Both chemotherapy and cetuximab function as radiosensitizers, meaning they can enhance the effectiveness of radiation therapy by increasing the tumor's sensitivity to radiation.
This is an important aspect because it suggests that, from a treatment mechanism perspective, both regimens should have a similar effect on the tumor microenvironment, potentially leading to similar radiological outcomes, especially in terms of MRI features. As you rightly pointed out, since both therapies serve this radiosensitizing role, we would not expect to see significant differences in the MRI sequences when comparing these two subgroups. Furthermore, our analysis did not show any significant variation between the MRI findings in patients receiving chemotherapy versus those receiving cetuximab, further supporting the hypothesis that both treatments operate in a similar manner in terms of their radiosensitizing effects. This is likely explained by the fact that MRI features primarily reflect changes in tumor biology, such as size, necrosis, or perfusion, which might not be dramatically influenced by the radiosensitizer itself but rather by the cumulative effect of radiation. We appreciate the reviewer’s valuable suggestion and hope that he/she agrees with our reasoning.
Lines 89-91: Is that very close follow-up is the standard in the whole country?
This is indeed the standard protocol according to both national and European guidelines (Clinical Practice Guidelines on Head and Neck Cancers). The rationale for the close follow-up is that head and neck tumors typically recur within the first few years after treatment, therefore, intensive follow-up is advised.
The features that You chose to analyze in image data analyses, why did You choose all of them? And further on that: in lines 222-225 You describe the differences of ADC values. Can there be a cut-off or a specific threshold of prognostic ADC value for all OPC patients? That is a general question.
Regarding the selection of radiomic features, we focused exclusively on features extracted from ADC maps, as our primary interest was in diffusion-related parameters and their potential relationship with HPV status. The selection of 105 features was a deliberate decision to find a balance between capturing a comprehensive range of radiomic features and maintaining statistical feasibility.
On the question of ADC cut-offs, we acknowledge that a universally applicable threshold for OPC prognosis has yet to be established. This is consistent with the findings in the literature, where ADC values show variability depending on factors such as tumor location, stage, and histological characteristics, and imaging parameters used in different studies. As a consequence, the ADC ranges among different studies are currently very large.
In another study of our research group, that is currently under revision in another journal, we have tried to identify a cut-off ADC value. However, due to the small number of events, especially in HPV positive disease, we were not able to perform this in a statistical correct manner. Furthermore, although low ADC values are generally associated with a higher likelihood of recurrence, some studies have not supported this correlation. In general, ADC values < 80 x 10-5 are considered to be associated with poor prognosis and higher risk for recurrence, while ADC values higher than 1000 are typically seen in less aggressive tumors. We believe that further prospective studies with larger patient cohorts and standardized imaging protocols are necessary to validate any potential ADC cut-off values and to determine their clinical applicability in routine practice.
What would the sensitivity or specificity of ADC values be for treatment response for OPC? Maybe not from Your data, but do you have any knowledge on that?
As the reviewer pointed out, our study indeed did not specifically analyze the sensitivity or specificity of ADC values for treatment response in OPC. However, we can provide some insight based on existing literature. Regarding sensitivity and specificity, the values for ADC as a predictor of treatment response in OPC again vary depending on the study design, imaging parameters, and patient population. In many studies, ADC values have demonstrated a sensitivity for detecting early treatment response, ranging between 70 to 90 %, depending on the study and the time point at which the ADC was measured. The specificity of ADC values, on the other hand, in predicting treatment response, is generally moderate. This can likely be explained by the fact that, while a significant change in ADC is suggestive of treatment response, other factors such as inflammation, fibrosis, … can also affect ADC values. Therefore, specificity values typically are around 60-70%. It is important to note that ADC values should not be used alone to assess treatment response. They are most useful when combined with other clinical and imaging findings, such as PET/CT scans or clinical assessments, to improve the accuracy of treatment evaluation.
